# Oral Health-Related Quality of Life among Adolescents in the First 6 Months of Fixed Orthodontic Therapy

**DOI:** 10.3390/ijerph20237110

**Published:** 2023-11-25

**Authors:** Cláudia Gonçalves Fahd, Gustavo Garcia Castro, Ana Carla Souza Costa, Cyrene Piazera Silva Costa, Ceci Nunes Carvalho, Célia Regina Maio Pinzan-Vercelino, Meire Coelho Ferreira

**Affiliations:** 1Post-Graduation Program in Dentistry, Ceuma University, São Luís 65075-120, Brazil; claudiafahd@hotmail.com (C.G.F.); anacarla.scosta@yahoo.com.br (A.C.S.C.); cyrene.piazerra@ceuma.br (C.P.S.C.); ceci.carvalho@ceuma.br (C.N.C.); 2Dentistry Department, University Center Ingá, Rio de Janeiro 87035-510, Brazil; cepinzan@hotmail.com

**Keywords:** adolescents, quality of life, orthodontics, longitudinal study

## Abstract

Clarification on disabilities that may arise during orthodontic treatment allows patients to have more realistic expectations. This prospective study assessed the impact of fixed orthodontic therapy on adolescents’ quality of life over 6 months. A total of 78 adolescents aged 11–17 years were included. Quality of life was measured using the Child Perceptions Questionnaire (CPQ_11–14_, short form) at five moments: before treatment (T0), one week (T1), one month (T2), three months (T3), and six months (T4) after treatment initiation. Multiple and pairwise comparisons were conducted for CPQ_11–14_ scores (Friedman and Wilcoxon test; effect size). Changes in the quality of life were assessed as mean differences (T0–T1 and T0–T4) in total and domain scores (Kruskal–Wallis and Mann–Whitney test) (α = 5%). Significant differences were observed between T0 and T4 in the oral symptoms’ domain (*p* < 0.001), and between T0 and T1, T2, T3, and T4 for emotional well-being (*p* < 0.001 for all). Significant differences in impact were also found between T0 and T2, T3, and T4 with regard to social well-being (*p* = 0.004, =0.049, and <0.001, respectively). Orthodontic therapy positively impacted the emotional and social aspects of adolescents’ quality of life. Negative effects were primarily related to pain, mouth sores, and difficulty biting or chewing. Understanding the symptoms and feelings of orthodontic patients aids professionals in decision-making.

## 1. Introduction

The primary objective of orthodontic therapy is to correct the occlusal function of the teeth and improve dental appearance, which is recognized as the main reason for seeking such therapy [1,2,3]. Considering the patient’s perspective to be crucial to the understanding of oral health needs, the World Health Organization suggests that quality of life measures that address symptoms and feelings should be considered as these can assist clinicians in decision-making, the planning of public health actions, and the evaluation of the effects of therapy [4].

Orthodontic patients can experience negative effects, such as pain and functional disability, throughout the course of therapy but especially in the early phase [5]. To better understand the disabilities experienced and provide patients with more realistic expectations regarding orthodontic therapy, prospective studies have investigated the impact of treatment by assessing patients’ quality of life [5,6,7,8,9,10,11].

Few studies have explored the impact of orthodontic therapy on the daily lives of children and adolescents [7,8,12,13]. Furthermore, no studies were found that investigated whether the type of malocclusion treated affected the outcomes experienced, nor studies in which orthodontic therapy was carried out by the same professional. The study aimed to investigate the impact of the first six months of fixed orthodontic treatment on the quality of life of adolescents. The null hypothesis of the study was that the impact of orthodontic treatment on the quality of life of adolescents does not decrease over the course of therapy.

## 2. Materials and Methods

This study received approval from the human research ethics committee of Ceuma University (1.819.530/2016). All participants read and signed a statement of informed consent form prior to participation in the study.

A prospective study was conducted with 78 adolescents recruited from three private clinics in São Luís, Brazil. The adolescents were included in the study in consecutive order between 2016 and 2019. The inclusion criteria were adolescents between 11 and 17 years of age, with no history of fixed orthodontic treatment, and without other oral conditions that could compromise their quality of life. As exclusion criteria, the adolescents with cognitive and mental difficulties that would enable a clear understanding of the questions of the quality of life’s instrument were excluded.

The sample size required to compare the means of impact [total score on Child Perceptions Questionnaire (CPQ_11–14_)] between different evaluation times was determined. Considering a 95% confidence level, 80% power, a standard deviation of 15.8 [5], and a minimum difference of five points to be detected in the mean total score between evaluation times (before and after 6 months of treatment), it was determined that a minimum of 78 adolescents was required.

The short form of the Brazilian Portuguese version of CPQ_11–14_ was used [14]. This instrument measures the impact of oral abnormalities on the quality of life of children and adolescents. It is composed of 16 items and four domains: oral symptoms (OS), functional limitations (FL), emotional well-being (EW), and social well-being (SW). It measures the frequency of events in the previous three months and on a scale with five response options: 0 = never, 1 = once/twice, 2 = sometimes, 3 = often, and 4 = almost every day. The total score ranges from 0 to 64 points, with higher scores denoting a greater negative impact on the quality of life. The CPQ_11–14_ is a validated instrument for use in adolescents aged between 11 and 14 years.

The CPQ_11–14_ was answered by the adolescents in the waiting room and completed without the interference of the researcher. The questionnaire was answered on four separate occasions: prior to orthodontic treatment (T0), one week (T1), one month (T2), three months (T3), and six months (T4) after the onset of treatment. The questionnaire was always completed before a scheduled appointment. The brackets were bonded to the upper arch on the first appointment and the lower arch after one month of treatment. Parents/caregivers answered a questionnaire on the demographic and socioeconomic characteristics of the adolescents. The type of malocclusion was extracted from the orthodontic documentation of adolescents.

Fixed orthodontic therapy was performed by the same orthodontist for all participants with pre-adjusted appliances (Roth prescription, slot.022″) using the standard sequence of nickel-titanium wires: 0.012″, 0.014″, 0.016″, 0.018″, 0.020″, and 0.016″ x 0.022″. During the study, tooth alignment and leveling were performed, with no cases of interproximal enamel reduction.

The dependent variable in the study was the impact of orthodontic treatment on quality of life (total CPQ11-14 score and domain scores), and the independent variables were the evaluation times (baseline/T0, T1, T2, T3, and T4).

The Statistical Package for the Social Sciences (IBM SPSS, version 21.0, Armonk, New York, NY, USA) was used for the data analysis. The Friedman test was used for comparing the total CPQ_11–14_, domain, and item scores among evaluations and the Wilcoxon test was used for the pairwise comparisons of evaluation times. The effect size for the Wilcoxon test was calculated [15]. The change in quality of life was measured by the mean difference (T0 minus T1 and T0 minus T4) in the total and domain scores according to the demographic, socioeconomic, and clinical variables, with a greater difference in score indicating a reduction in the negative impact on quality of life. The Kruskal–Wallis and Mann–Whitney tests were used to compare the groups. The significance level was set to 5%.

## 3. Results

Table 1 shows the demographic, socioeconomic, and clinical findings of the sample. The mean age of the participants was 14.12 ± 1.95 years. Class I malocclusion was predominant (Table 1).

Significant differences were found among all evaluation times for the total CPQ_11–14_ and domain scores. In the pairwise comparisons, significant differences were found between T0 and other times for different domains (*p* < 0.05). A gradual, significant reduction in the mean scores was found for the EW and SW domains, and it was clinically evidenced by a large effect size between the baseline (T0) and 6 months (T4) from the start of treatment (d = 0.91) (Table 2).

The most important findings in Table 3 include the demonstration of an increase in the score for the item “pain in teeth” reported 1 week after bonding, followed by a gradual decrease. A gradual increase was also found in the score for the “difficulty biting or chewing” item, followed by a reduction in the score at the 6-month evaluation. A gradual reduction was found for the items “felt irritated or frustrated”, “felt shy, embarrassed or ashamed”, “was upset”, and “avoided smiling or laughing” (Table 3).

Significant changes in the quality of life regarding the SW domain were found for “self-declared skin color” (T0 minus T1, *p* = 0.041; T0 minus T4, *p* = 0.029). A significant change was also found regarding the “OS” domain for “family income” (T0 minus T1, *p* = 0.029) (Table 4).

## 4. Discussion

This study investigated the impact of six months of fixed orthodontic therapy on the quality of life of Brazilian adolescents. The null hypothesis of the study was partially rejected. The negative impact on EW and SW decreased over the first six months, demonstrating the positive effects of treatment, both in the way that the adolescents perceived their dental appearance and the greater acceptance by their peers, both of which are important aspects in the construction of a personal identity [16]. This is especially true for young Brazilians, who believe that orthodontic therapy results in better aesthetics and, consequently, better social relations [17].

The reduction in psychosocial impact is also related to the cooperation of the patient with treatment [8]. In the present study, all patients underwent treatment with the same orthodontist using the same therapeutic mechanics, and a care protocol was established that involved the patient’s education and motivation for orthodontic therapy. Such actions contribute to a lower frequency of complications and, consequently, translate to therapeutic success.

The reduction in the impact on EW is in agreement with findings from previous studies involving adolescents followed up for 1 month [7], 1 year after the start of orthodontic treatment, and 1 month after the end of treatment [12], demonstrating that orthodontic treatment is capable of reducing the negative psychological aspects. In addition, a study of adolescents showed that after 1 year of orthodontic therapy, there was a significant reduction in impact related to FL, EW, and SW [8]. These findings show that even the early phase of orthodontic therapy tends to contribute to a reduction in psychosocial impact, as reported in a longitudinal study conducted with Chinese adolescent orthodontic patients [11].

The SW domain reflects inadequate social behaviors due to subjective perceptions of an unfavorable dental appearance. For this domain, the adolescents reported a reduction in impact throughout the course of treatment for the items “argued with other children or people in the family” and “was teased or called names”. However, greater impact was found for the item “asked questions about your teeth”, which suggests greater interest and curiosity on the part of others regarding the orthodontic treatment.

In a study conducted with Chinese children using the long form of the CPQ, a significant increase in the impact of orthodontic treatment was found when comparing the evaluation prior to treatment to all other evaluation times (one month, three months, and six months) [5]. In the present study, a significant reduction in impact regarding the “OS” domain was found at the six-month evaluation period compared to the pre-treatment evaluation.

The reduction in the negative impact of orthodontic therapy after 6 months demonstrates that the initial negative effects tend to diminish in intensity over the course of treatment. According to a systematic review, impact scores increase in the initial weeks following the placement of the appliance but tend to diminish in the final stages of therapy [18].

Considering the items of the CPQ_11–14_, pain was more pronounced one week after the bonding of the brackets and decreased over the subsequent evaluation times (1, 3, and 6 months). This finding is in agreement with data from a longitudinal study conducted with Chinese adolescents, in which pain was more evident 1 week after the bonding of the brackets and diminished progressively throughout the course of treatment [6]. In another study that evaluated pain after the placement of two initial wires of different sizes (0.014 and 0.016 inches), pain initiated 2 h after the placement of the wires, affecting 32% of the group with 0.014” wires and 35.7% of the group with 0.016” wires. After 6 h of applied use, the frequency of pain rose to 83.9% and 88.1%, respectively. On the seventh day, the proportion of patients impacted by pain dropped to 41% and 26.4%, respectively [19].

Orthodontic pain is commonly attributed to tooth discomfort induced by the imposed force. However, it also refers to pain from ulcers in the mucosa, tongue discomfort, and lesions on the gingival tissue [20]. Pain stemming from the force applied to the teeth induces an inflammatory response in the surrounding tissues [21]. Therefore, it is necessary to minimize the pain process so that treatment can progress and oral hygiene can be performed effectively. To curb pain, pharmacological, behavioral, and low-level laser therapies may be necessary [21,22].

The gradual increase in impact regarding the item “mouth sores” occurs due to trauma to the oral mucosa (cheeks, lips, and tongue) caused by the orthodontic brackets [23]. The installation of parts with hooks and sharp corners causes the previously smooth mucosa to experience friction and exhibit the formation of sores [24]. Moreover, initially uncompromised areas of mucosa become affected by the progression of tooth movement.

The retention of food scraps is common for crowded teeth and tends to worsen after the placement of an orthodontic appliance. In the present study, the item “food caught between teeth” did not worsen after the installation of an appliance, but continued compromising the daily lives of adolescents up to the 6-month evaluation.

The types of malocclusions have different dental and bone characteristics; therefore, the authors speculated that changes in the quality of life could be different, especially during initial dental alignment and leveling. However, the results showed that the type of malocclusion did not influence the impact stemming from OS at the first week of treatment.

“Difficulty biting or chewing” exerted the same impact from one week to three months after the bonding of the brackets. This finding was expected, as both the pain experienced and the limitation imposed by the orthodontic appliance tend to compromise chewing function [9,25]. Painful teeth with mobility make it uncomfortable for the patients to bite and chew stiff and fibrous foods [25]. However, this difficulty tends to improve with the evolution of treatment [26], as seen at the 6-month evaluation in the present study.

In the comparison of the pre-treatment and six-month evaluations, malocclusion was found to exert an influence on the impact of FL. The lower average change in the quality of life found for individuals with Class I and III malocclusions demonstrates a smaller reduction in impact from T0 to T4. The highest average change in the quality of life found for Class II individuals was probably due to this type of malocclusion being the major reason patients seek orthodontic treatment [27]. Orthodontic treatment is often undertaken due to an aesthetic impairment that, with treatment, is resolved, helping patients feel better.

The lower negative impact for the item “felt shy, embarrassed or ashamed” may be related to the fact that the adolescents did not consider themselves to be strange to their peers, as the use of an orthodontic appliance is commonplace among adolescents. The social inclusion of being accepted by one’s peers tends to impede the establishment of conflicts, which is confirmed by the fact that the item “argued with other children or people in the family” contributed least to the negative impact on quality of life throughout orthodontic therapy.

Regarding the change in the quality of life related to the “SW” domain, the only significant difference was related to self-declared race. In the comparison of the pre-treatment and 1-week evaluations, the greatest change was found for the black race, suggesting that even after a short treatment period, these adolescents tended to perceive orthodontic treatment as advantageous because the promise of aesthetic improvement is capable of boosting one’s self-confidence and feelings of social inclusion. The same occurred in the comparison of the pre-treatment and 6-month evaluations for both the black and brown races.

Considering the methodological aspects of the study, the CPQ_11–14_ was administered during treatment, which enabled the respondents to offer a reliable report of symptoms and feelings. The extension of the CPQ_11–14_ to adolescents aged 15 to 17 years is considered valid because the 10- to 19-year-old age group is classified by the World Health Organization as adolescence. Moreover, the short form of the CPQ_11–14_ is capable of detecting higher levels of impact on the quality of life than the long version because the questions in the short form address the main problems experienced by adolescents [28], especially those undergoing orthodontic treatment. It is also important to highlight the reliability of the findings as a result of the eligibility criteria, such as the non-inclusion of adolescents with other oral conditions that could compromise their quality of life. It should also be noted that 74.4% of the participants presented with Class I malocclusion and that all study participants underwent orthodontic therapy by the same professional. A limitation of this study is that the convenience sample is considered, which prevents the extrapolation of the results to populations of adolescents undergoing orthodontic treatment. Another limitation is that when performing a subjective assessment through a quality instrument, self-reports may not capture the full range of experiences and emotions related to orthodontic treatment.

Because orthodontic treatment is a complex intervention, understanding the symptoms and feelings of orthodontic patients aids professionals in decision-making. The orthodontist’s instructions regarding the consequences and periods of greater discomfort during treatment give the patients prior knowledge of the symptoms that may arise and enable more realistic expectations of the therapy [6]. Evaluating the oral-health-related quality of life in patients contributes to the practice of orthodontics.

The present findings provide useful information on the first six months of orthodontic therapy, showing that the psychosocial impact tends to diminish throughout the course of treatment, whereas symptoms and FL remain similar. Thus, young patients should be warned at the onset of therapy that the greatest discomfort occurs in the first few months of using a fixed orthodontic appliance.

Based on the findings for adolescents, future research could focus on adult populations starting orthodontic treatment. Given that therapy duration is generally longer due to the extended time required for periodontal remodeling, it is expected that the impacts on the quality of life of patients could be more significant.

## 5. Conclusions

Orthodontic treatment was found to have a positive emotional and social impact on the quality of life of adolescents. The negative aspects were mainly related to pain, mouth sores, food caught between the teeth, and difficulties in biting and chewing.

## Figures and Tables

**Table 1 ijerph-20-07110-t001:** Demographic, socioeconomic, and clinical characteristics of sample (*n* = 78).

	*n* (%)
Sex	
Male	37 (47.4)
Female	41 (52.6)
Age group	
11 to 14 years	46 (59)
15 to 17 years	32 (41)
Self-declared skin color	
Brown	45 (57.7)
Black	16 (20.5)
White	17 (21.8)
Caregiver’s schooling	
<8 years of study	23 (29.5)
≥8 years of study	55 (70.5)
Family income *	
<2 BMMW	33 (42.3)
2 to <5 BMMW	34 (43.6)
5 to <10 BMMW	9 (11.5)
Malocclusion	
Class I	58 (74.4)
Class II	15 (19.2)
Class III	5 (6.4)

BMW: Brazilian monthly minimum wage (Approximately U$ 250); * Missing data.

**Table 2 ijerph-20-07110-t002:** Mean score of total CPQ_11–14_ and domains prior to and after different times of treatment.

CPQ_11–14_	T0 Mean (SD)	T1 Mean (SD)	T2 Mean (SD)	T3 Mean (SD)	T4 Mean (SD)	*p* *	*p* ** d	*p* ^§^ d	*p* ^¥^ d	*p* ^†^ d
TOTAL SCORE	12.23 (8.2)	11.40 (6.5)	10.72 (6.3)	10.62 (6.4)	6.82 (6.2)	<0.001	0.305 0.11	0.093 0.21	0.265 0.22	<0.001 0.75
OS	3.94 (1.9)	4.05 (1.9)	4.23 (2.2)	4.32 (2.2)	2.86 (2.4)	<0.001	0.558 −0.06	0.340 −0.14	0.155 −0.18	0.002 0.50
FL	2.49 (2.3)	3.06 (2.6)	3.19 (2.5)	3.19 (2.4)	2.08 (2.0)	0.027	0.095 −0.23	0.058 −0.29	0.055 −0.30	0.272 0.19
EW	3.29 (3.6)	2.24 (2.6)	1.53 (1.9)	1.42 (1.8)	0.74 (1.7)	<0.001	<0.001 0.33	<0.001 0.61	<0.001 0.66	<0.001 0.91
SW	2.51 (2.8)	2.04 (1.9)	1.77 (1.9)	1.68 (1.9)	1.14 (1.9)	<0.001	0.115 0.20	0.004 0.31	0.049 0.35	<0.001 0.57

SD: standard deviation; * comparison of mean impact among all evaluation times; ** comparison of mean impact between T0 and T1; ^§^ comparison of mean impact between T0 and T2; ^¥^ comparison of mean impact between T0 and T3; ^†^ comparison of mean impact between T0 and T4; effect size (d): d ≤ 0.20 (small), d from >0.20 to <0.80 (moderate), d ≥ 0.80 (large) [15].

**Table 3 ijerph-20-07110-t003:** Severity of impact on CPQ_11–14_ items prior to bonding of orthodontic brackets (T0) and after one week (T1), one month (T2), three months (T3), and six months (T4).

Domains/Items CPQ_11–14_	T0	T1	T2	T3	T4	
	Mean (SD)	Mean (SD)	Mean (SD)	Mean (SD)	Mean (SD)	*p* *
OS						
Pain in teeth, lips, jaws or mouth	0.63 (0.81)	1.03 (0.90)	0.91 (0.84)	0.86 (0.70)	0.28 (0.62)	<0.001
Mouth sores	0.40 (0.67)	0.58 (0.68)	0.95 (0.74)	1.03 (0.93)	0.69 (0.76)	<0.001
Bad breath	1.15 (1.13)	0.85 (0.85)	0.68 (0.81)	0.69 (0.76)	0.72 (0.84)	0.002
Food caught between teeth	1.76 (0.91)	1.60 (0.81)	1.69 (0.90)	1.74 (0.95)	1.17 (1.06)	<0.001
FL						
Taken longer to eat meals	0.60 (0.87)	0.67 (0.91)	0.68 (0.88)	0.67 (0.86)	0.33 (0.73)	0.009
Difficulty biting or chewing	0.51 (0.95)	1.06 (1.30)	1.14 (1.04)	1.12 (1.07)	0.60 (0.76)	<0.001
Difficulty speaking	0.37 (0.82)	0.32 (0.61)	0.45 (0.78)	0.44 (0.73)	0.33 (0.66)	0.286
Difficulty drinking or eating hot/cold foods	1.00 (1.11)	1.01 (1.10)	0.92 (1.09)	0.97 (1.15)	0.78 (1.08)	0.545
EW						
Felt irritated or frustrated	0.64 (0.90)	0.53 (0.79)	0.38 (0.78)	0.37 (0.76)	0.13 (0.47)	<0.001
Felt shy, embarrassed or ashamed	0.87 (1.04)	0.51 (0.80)	0.37 (0.71)	0.37 (0.63)	0.32 (0.71)	<0.001
Was upset	0.59 (0.87)	0.42 (0.68)	0.36 (0.66)	0.18 (0.48)	0.09 (0.33)	<0.001
Concerned about what others think of your teeth, lips or jaws	1.19 (1.36)	0.78 (0.98)	0.41 (0.69)	0.50 (0.83)	0.21 (0.57)	<0.001
SW						
Avoided smiling or laughing	0.83 (1.21)	0.74 (1.10)	0.51 (1.03)	0.45 (0.98)	0.44 (1.04)	0.001
Argued with other children or people in the family	0.33 (0.89)	0.14 (0.39)	0.18 (0.42)	0.18 (0.48)	0.15 (0.43)	0.960
Was teased or called names	0.63 (1.12)	0.38 (0.78)	0.40 (0.69)	0.33 (0.70)	0.15 (0.65)	0.002
Asked questions about your teeth	0.72 (0.88)	0.77 (0.94)	0.68 (0.90)	0.72 (0.87)	0.40 (0.69)	0.001

Friedman test. * *p* < 0.05.

**Table 4 ijerph-20-07110-t004:** Changes in quality of life based on mean difference in domains and total CPQ_11–14_ according to demographic, socioeconomic, and clinical variables.

	Change in Quality of Life									
Domains/Total Score	OS	FL	EW	SW	Total Score	OS	FL	EW	SW	Total Score
	T0–T1					T0–T4				
	Mean (SD)	Mean (SD)	Mean (SD)	Mean (SD)	Mean (SD)	Mean (SD)	Mean (SD)	Mean (SD)	Mean (SD)	Mean (SD)
Sex										
Male	1.34 (1.11)	2.03 (1.93)	1.80 (2.26)	0.97 (1.65)	4.43 (3.55)	2.66 (1.64)	1.94 (1.91)	2.83 (3.24)	2.37 (2.43)	8.26 (7.42)
Female	1.54 (1.33)	2.10 (2.17)	1.44 (1.71)	1.49 (2.22)	5.15 (4.62)	2.56 (1.96)	2.20 (2.36)	3.05 (2.94)	2.10 (2.66)	8.15 (7.45)
Age group										
11 to 14 years	1.48 (1.25)	2.11 (2.29)	1.48 (1.81)	1.30 (2.31)	5.14 (4.69)	2.80 (1.88)	2.48 (2.34)	2.50 (3.03)	2.45 (2.92)	8.59 (8,48)
15 to 17 years	1.41 (1.21)	2.0 (1.70)	1.78 (2.21)	1.19 (1.44)	4.38 (3.29)	2.34 (1.72)	1.53 (1.76)	3.56 (3.05)	1.91 (1.90)	7.66 (5.63)
Self-declared skin color										
Brown	1.20 (1.00)	2.14 (2.26)	1.52 (1.94)	1.18 (2.08) *	4.41 (4.39)	2.50 (1.91)	1.91 (2.23)	3.50 (3.09)	2.59 (2.67) *	9.23 (7.48)
Black	2.06 (1.53)	1.75 (1.61)	1.88 (2.36)	2.00 (2.09)	5.94 (3.47)	3.0 (2.03)	2.13 (2.03)	2.69 (3.18)	2.56 (2.68)	8.38 (8.37)
White	1.50 (1.31)	2.19 (1.91)	1.56 (1.75)	0.69 (1.40)	4.81 (4.14)	2.50 (1.27)	2.50 (2.10)	1.69 (2.60)	0.88 (1.50)	5.19 (5.40)
Caregiver’s schooling										
<8 years of study	1.87 (1.39)	2.09 (2.02)	1.74 (2.13)	1.22 (1.90)	5.17 (4.42)	2.22 (1.81)	2.0 (1.54)	2.48 (2.78)	1.70 (2.25)	6.22 (6.80)
≥8 years of study	1.26 (1.11)	2.06 (2.08)	1.55 (1.92)	1.26 (2.04)	4.66 (4.07)	2.77 (1.80)	2.11 (2.38)	3.15 (3.18)	2.45 (2.65)	9.06 (7.52)
Family income ^¥^										
<2 × BMMW	1.82 (1.38) *	2.21 (2.27)	1.42 (1.92)	1.33 (1.81)	5.27 (3.92)	2.58 (1.99)	2.30 (2.11)	2.91 (2.90)	2.27 (2.04)	8.06 (6.90)
2 to <5 × BMMW	1.26 (1.11)	2.24 (2.00)	1.76 (2.07)	1.24 (2.28)	4.74 (4.61)	2.56 (1.76)	1.71 (2.14)	2.88 (3.15)	1.94 (2.81)	7.97 (7.56)
5 to <10 × BMMW	0.78 (0.44)	0.89 (0.78)	1.67 (2.00)	1.00 (1.50)	3.44 (3.09)	2.89 (1.45)	2.67 (2.34)	3.33 (3.60)	3.11 (3.18)	9.56 (9.12)
Malocclusion										
Class I	1.33 (1.08)	1.89 (1.97)	1.51 (1.97)	0.98 (1.49)	4.00 (2.76)	2.56 (1.82)	1.79 (1.94) *	2.88 (3.03)	1.93 (2.21)	7.68 (6.68)
Class II	1.93 (1.77)	2.93 (2.43)	2.21 (1.93)	2.07 (2.89)	8.29 (6.74)	2.86 (1.75)	3.36 (2.65)	3.29 (3.02)	3.07 (3.10)	10.14 (8.67)
Class III	1.40 (0.89)	1.60 (1.52)	1.00 (2.23)	2.00 (3.39)	4.40 (4.16)	2.40 (2.19)	1.80 (2.05)	2.80 (4.08)	3.20 (4.08)	8.60 (11.05)

T0–T1: prior to onset of treatment minus one week after bonding of orthodontic brackets; T0–T4: prior to onset of treatment minus six months after bonding of orthodontic brackets; ^¥^ missing data; * *p* < 0.05; Kruskal–Wallis test; Mann–Whitney test.

## Data Availability

The data presented in this study are available on request from the corresponding author. The data are not publicly available due to [restrictions, e.g., privacy or ethical].

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
