# Peer review of "Oral Health-Related Quality of Life among Adolescents in the First 6 Months of Fixed Orthodontic Therapy"

_ijerph, 2023, doi:10.3390/ijerph20237110_

Round 1
Reviewer 1 Report
Comments and Suggestions for Authors
The manuscript titled "Impact of Fixed Orthodontic Therapy on Adolescents' Quality of Life Over Six Months" explores the influence of orthodontic treatment on adolescents' quality of life. The study is generally well-structured and addresses an important topic. However, some aspects require critical review and consideration:
- Abstract:
- The abstract is clear and concise, providing a good overview of the study's objectives, methods, and key findings.
- It would be helpful to provide some additional context or background information about the significance of studying the impact of orthodontic therapy on adolescents' quality of life. Why is this an important topic, and what are the potential implications of the findings? This would help readers understand the broader relevance of the study.
- Consider providing a sentence at the end of the abstract summarizing the practical implications of the study's findings. How might this research benefit orthodontic treatment planning or the well-being of adolescents undergoing such treatment?
2. Introduction:
- The introduction provides a clear and concise overview of the study's objectives and the relevance of orthodontic therapy to oral health. It effectively communicates the primary objective of orthodontic therapy.
- It would be helpful to include a brief statement or hypothesis that outlines what the study expects to find concerning the impact of the first six months of fixed orthodontic treatment on adolescents' quality of life. This would give readers a sense of the study's direction and objectives from the outset.
3. Materials and Methods:
- The materials and methods section of your study is well-structured and provides a clear overview of the research design and procedures. The description of the prospective study design is clear and concise, providing essential details about the study population, location, and inclusion/exclusion criteria. The description of the statistical analysis tools and methods used is clear and understandable.
- While the description of the CPQ11-14 is good, you might consider briefly explaining why this specific questionnaire was chosen. What led you to select this tool for your study over others? Is it widely recognized or validated?
- The instruments used for data collection (CPQ11-14) and the evaluation times are well-explained, but there is limited information on how the questions were administered, which is important to avoid response bias.
- While the study received ethical approval, more information on the informed consent process and the handling of sensitive data would be valuable.
4. Results:
- The Results section effectively presents the key findings of the study, discussing the demographic characteristics of the participants, significant changes in CPQ11-14 scores over time, and the impact of specific items on the questionnaire.
- The presentation of results in tables is clear and informative. However, the text explanation of the results is somewhat repetitive and could be more concise.
- Consider adding a brief title above the table to describe its content (e.g., "Table 2: Impact on CPQ11-14 Domains Before and After Orthodontic Treatment").
5. Discussion:
- The discussion section of the research paper effectively addresses the impact of orthodontic treatment on the quality of life of Brazilian adolescents. It provides a balanced overview of orthodontic treatment's positive and negative effects while discussing relevant methodological aspects.
- While the discussion mentions a reduction in negative impact, it lacks specific quantitative information or effect sizes. Providing numerical data or effect size measures would help readers better understand the significance of the observed changes.
- Including a brief subsection or sentences on potential areas for future research would be beneficial. This could stimulate further investigations and discussions in the orthodontic community.
- The study's limitations and potential sources of bias should be acknowledged. The discussion acknowledges that the convenience sample limits the generalizability of the results. However, it's important to recognize that self-report questionnaires may not capture the full range of experiences and emotions related to orthodontic treatment.
- The discussion acknowledges that the convenience sample limits the generalizability of the results but does not elaborate on the potential implications of this limitation.
In summary, the manuscript addresses an important research question and presents valuable findings regarding the impact of orthodontic treatment on adolescents' quality of life. However, there is room for improvement regarding organization, clarity, depth of analysis, and a more comprehensive discussion of the results in the context of existing literature.
Comments on the Quality of English LanguageThe manuscript is well-structured, maintains a formal tone, and effectively communicates complex research findings. The English in the manuscript is strong, with only minor suggestions for improvement in sentence structure, word choice, and brevity. The clarity and formality of the writing contribute to the document's quality and readability.
Author Response
Comments and Suggestions for Authors
The manuscript titled "Impact of Fixed Orthodontic Therapy on Adolescents' Quality of Life Over Six Months" explores the influence of orthodontic treatment on adolescents' quality of life. The study is generally well-structured and addresses an important topic. However, some aspects require critical review and consideration:
- Abstract:
- The abstract is clear and concise, providing a good overview of the study's objectives, methods, and key findings.
- It would be helpful to provide some additional context or background information about the significance of studying the impact of orthodontic therapy on adolescents' quality of life. Why is this an important topic, and what are the potential implications of the findings? This would help readers understand the broader relevance of the study.
- Consider providing a sentence at the end of the abstract summarizing the practical implications of the study's findings. How might this research benefit orthodontic treatment planning or the well-being of adolescents undergoing such treatment?
Authors’ response: In response to the 2nd topic, background information about the significance of study was inserted at the end of the abstract.
- Introduction:
- The introduction provides a clear and concise overview of the study's objectives and the relevance of orthodontic therapy to oral health. It effectively communicates the primary objective of orthodontic therapy.
- It would be helpful to include a brief statement or hypothesis that outlines what the study expects to find concerning the impact of the first six months of fixed orthodontic treatment on adolescents' quality of life. This would give readers a sense of the study's direction and objectives from the outset.
Authors’ response: In response to the 2nd topic, the study's null hypothesis has been added to the last paragraph of the introduction.
- Materials and Methods:
- The materials and methods section of your study is well-structured and provides a clear overview of the research design and procedures. The description of the prospective study design is clear and concise, providing essential details about the study population, location, and inclusion/exclusion criteria. The description of the statistical analysis tools and methods used is clear and understandable.
- While the description of the CPQ11-14 is good, you might consider briefly explaining why this specific questionnaire was chosen. What led you to select this tool for your study over others? Is it widely recognized or validated?
Authors’ response: Informations about the CPQ11-14 were inserted: “Brazilian portuguese version…" and “The CPQ11-14 is a validated instrument for adolescents aged 11 to 14." (4th paragraph of Materials and Methods section).
- The instruments used for data collection (CPQ11-14) and the evaluation times are well-explained, but there is limited information on how the questions were administered, which is important to avoid response bias.
Authors’ response: The CPQ11-14 was answered by the adolescents. This information has been added to the 5th paragraph of the Material and Methods section.
- While the study received ethical approval, more information on the informed consent process and the handling of sensitive data would be valuable.
Authors’ response: Information about the Informed Consent Form has been added to the 1st paragraph of the Materials and Methods section: “All participants read and signed a statement of informed consent form prior to participation in the study.”
- Results:
- The Results section effectively presents the key findings of the study, discussing the demographic characteristics of the participants, significant changes in CPQ11-14 scores over time, and the impact of specific items on the questionnaire.
- The presentation of results in tables is clear and informative. However, the text explanation of the results is somewhat repetitive and could be more concise.
Authors’ response: In order to make the text of the Results section more concise, the following exclusions were made:
It has been deleted from the 1st paragraph: “and 42.3% of the adolescents’ families had an income less than two times the Brazilian monthly minimum wage.”
It has been deleted from the 2nd paragraph: “...T4 for the “OS” domain, between T0 and T1, T2, T3, and T4 for the “EW” domain, and between T0 and T2, T3, and T4 for the “SW” domain.”
It has been deleted from the 3rd paragraph: “...For the item “mouth sores”, a gradual increase was found in the mean score from one week after bonding, and a reduction was observed at the six-month evaluation....”; “…“concerned about what others think”…”
It has been deleted from the 4th paragraph: “…and a significant change was found regarding the "FL" domain for "malocclusion" (T0 minus T4, p=0.048).”
- Consider adding a brief title above the table to describe its content (e.g., "Table 2: Impact on CPQ11-14 Domains Before and After Orthodontic Treatment").
Authors’ response: Thank you for your suggestion. The statement in table 2 has been modified to: “Mean score of total CPQ11-14 and domains prior and after different times of treatment”.
- Discussion:
- The discussion section of the research paper effectively addresses the impact of orthodontic treatment on the quality of life of Brazilian adolescents. It provides a balanced overview of orthodontic treatment's positive and negative effects while discussing relevant methodological aspects.
- While the discussion mentions a reduction in negative impact, it lacks specific quantitative information or effect sizes. Providing numerical data or effect size measures would help readers better understand the significance of the observed changes.
Authors’ response: Thank you for your suggestion. In order to help interpret the significance of the findings in terms of reducing negative impacts on quality of life, the effect size was calculated between the initial time (baseline) and the other evaluation times (1 week, 1 month, 3 and 6 months).
- Including a brief subsection or sentences on potential areas for future research would be beneficial. This could stimulate further investigations and discussions in the orthodontic community.
Authors’ response: Thank you for your suggestion. A paragraph has been added to the end of the discussion in which we report on what we consider important to investigate in future research.
- The study's limitations and potential sources of bias should be acknowledged. The discussion acknowledges that the convenience sample limits the generalizability of the results. However, it's important to recognize that self-report questionnaires may not capture the full range of experiences and emotions related to orthodontic treatment.
Authors’ response: The information about “self-report questionnaires may not capture the full range of experiences and emotions related to orthodontic treatment” has been added to the paragraph that talks about the strengths and limitations of the study (16th paragraph of the Discussion section).
- The discussion acknowledges that the convenience sample limits the generalizability of the results but does not elaborate on the potential implications of this limitation.
Authors’ response: The implications of the convenience sample have been added in the paragraph that talks about the strengths and limitations of the study (16th paragraph of the Discussion section).
In summary, the manuscript addresses an important research question and presents valuable findings regarding the impact of orthodontic treatment on adolescents' quality of life. However, there is room for improvement regarding organization, clarity, depth of analysis, and a more comprehensive discussion of the results in the context of existing literature.
Comments on the Quality of English Language
The manuscript is well-structured, maintains a formal tone, and effectively communicates complex research findings. The English in the manuscript is strong, with only minor suggestions for improvement in sentence structure, word choice, and brevity. The clarity and formality of the writing contribute to the document's quality and readability.
Authors’ response: The English language was revised (Text in purple color).
Based on reviewer 1's suggestions, all changes/additions made to the text of the article are in red font.

Reviewer 2 Report
Comments and Suggestions for Authors
This paper purposed to investigate the impact of the first six months of fixed orthodontic treatment on the quality of life of adolescents. I do have some comments as listed below in the order noted.
Comment 1:
What is the novelty of this study although Raji Alrwuili M et al. currently published one article, entitled “A Detailed Correlation of Oral-Health-Related Quality of Life of Patients Undergoing Fixed Orthodontic Therapy” in the Cureus 2023;15(1):e33854?
Comment 2:
In Introduction, the authors claim that studies have explored the impact of orthodontic therapy on the daily lives of children and adolescents. The authors should cite the references Sauer MK et al., Clin Oral Investig 2023;27(1):369-375; Raji Alrwuili M et al., Cureus 2023;15(1):e33854; and Freitas LRP & Oliveira DD, Dental Press J Orthod 2021;26(5):e21bbo5, which were to examine the oral health-related quality of life and oral hygiene in adolescents before and during aligner therapy.
Comment 3:
Please add a paragraph about the contribution/significance of this article in a bulleted form at the end part of the Introduction section.
Comment 4:
The quality of the data set is very important, especially for 78 adolescents recruited from three private clinics in São Luís, Brazil. For this reason, please clarify the inclusion criteria and exclusion criteria of sample collection in the Materials and Methods section and please also provide a flowchart immediately after this section.
Comment 5:
Authors should describe more details of how to choose the variables used in his study.
Comment 6:
Please provide the Strengths of the study in the Discussion section.
Author Response
Comments and Suggestions for Authors
This paper purposed to investigate the impact of the first six months of fixed orthodontic treatment on the quality of life of adolescents. I do have some comments as listed below in the order noted.
Comment 1: What is the novelty of this study although Raji Alrwuili M et al. currently published one article, entitled “A Detailed Correlation of Oral-Health-Related Quality of Life of Patients Undergoing Fixed Orthodontic Therapy” in the Cureus 2023;15(1):e33854?
Authors’ response: The difference in our study is that it was carried out with a group of patients treated by the same orthodontist. This information was included in the last paragraph of the Introduction section.
Comment 2: In Introduction, the authors claim that studies have explored the impact of orthodontic therapy on the daily lives of children and adolescents. The authors should cite the references Sauer MK et al., Clin Oral Investig 2023;27(1):369-375; Raji Alrwuili M et al., Cureus 2023;15(1):e33854; and Freitas LRP & Oliveira DD, Dental Press J Orthod 2021;26(5):e21bbo5, which were to examine the oral health-related quality of life and oral hygiene in adolescents before and during aligner therapy.
Authors’ response: The study of Raji et al. (2023) was not cited because it is a narrative review, without a systematic search of the literature, nor was assessed the quality of the primary studies included. In addition, most of the primary studies cited in this review used the OHIP, a quality of life’s instrument aimed at an age group other than the present study.
The study of Sauer et al. (2023) was not cited because the quality of life’s instrument used is different from the one used in this study, and the alignment system used to treat the patients is different from the one used in the sample of people in our study.
The study of Freitas and Oliveira (2021) dealt with two clinical cases, one of a child patient (9.8 year-old) and the other of a young adult patient (21 year-old), both of whom had undergone orthodontic retreatment, unlike our study, which dealt with adolescents who had never undergone orthodontic treatment. We believe that patients who have already undergone orthodontic treatment have fewer limitations in terms of oral health-related quality of life, since they have already experienced treatment in the past and are therefore able to cope with the impacts on daily life in a different way to patients who have undergone orthodontic therapy for the first time.
Comment 3: Please add a paragraph about the contribution/significance of this article in a bulleted form at the end part of the Introduction section.
Authors’ response: Thank you for your suggestion. As the journal's rules do not accept "bulleted points", nor at the end part of the Introduction section, the contribution of the study was inserted in the penultimate paragraph of the Discussion section.
Comment 4: The quality of the data set is very important, especially for 78 adolescents recruited from three private clinics in São Luís, Brazil. For this reason, please clarify the inclusion criteria and exclusion criteria of sample collection in the Materials and Methods section and please also provide a flowchart immediately after this section.
Authors’ response: Information on the study's inclusion and exclusion criteria has been clarified in the text (2nd paragraph of Materials and Methods section).
Comment 5: Authors should describe more details of how to choose the variables used in his study.
Authors’ response: The variables explored in the study, in order to answer the objective of the study, have been inserted in Materials and Methods (penultimate paragraph of Materials and Methods section).
Comment 6: Please provide the Strengths of the study in the Discussion section.
Authors’ response: The strengths of the study are highlighted in the 16th paragraph of the Discussion section, adding to the text: "It should also be noted that 74.4% of the participants had class I malocclusion and that all the participants in the study underwent orthodontic therapy by the same professional."
Based on reviewer 2's suggestions, all changes/additions made to the text of the article are in blue font.

Reviewer 3 Report
Comments and Suggestions for Authors
COMMENTS TO THE AUTHORS
Dear editors and researchers, I was happy to evaluate your article. However, I believe that making the revisions I have mentioned below will make a positive contribution to your article:
ABSTRACT
1. Revise the expression (T0 minus T1 and T0 minus T4) in the Abstract as (T0-T1 and T0-T4).
INTRODUCTION
1. ‘However, ethnic 37 differences often lead to different oral beliefs and behaviors [12,13].’ The relevant sentence has no relation with the article. It would be an appropriate decision to remove it.
2. ‘Considering 41 the negative effects of orthodontic therapy in the first months of treatment and the need 42 to strengthen the available evidence on this issue, the study aimed to investigate the 43 impact of the first six months of fixed orthodontic treatment on the quality of life of 44 adolescents.’ The understandability of the relevant sentence is low and may cause confusion in the reader. My advice to researchers is to divide the sentence and present the purpose of the study clearly to the reader as a separate sentence.
3. Please add the null hypothesis sentence after the sentence explaining the purpose of the study and make a statement in the discussion section that you accept or reject your null hypothesis based on your results.
RESULTS
1. There are too many tables in the results section and it negatively affects the understandability of the article. Researchers do not need to describe all findings one by one. They can highlight important findings and reference others in the table.
GENERAL COMMENTS
I believe that the article needs serious editing in terms of writing language. Please have the entire text revised professionally.
Comments on the Quality of English LanguageI believe that the article needs serious editing in terms of writing language. Please have the entire text revised professionally.
Author Response
Comments and Suggestions for Authors
Dear editors and researchers, I was happy to evaluate your article. However, I believe that making the revisions I have mentioned below will make a positive contribution to your article:
ABSTRACT
- Revise the expression (T0 minus T1 and T0 minus T4) in the Abstract as (T0-T1 and T0-T4).
Authors’ response: The expression was revised in the abstract.
INTRODUCTION
- ‘However, ethnic differences often lead to different oral beliefs and behaviors [12,13].’ The relevant sentence has no relation with the article. It would be an appropriate decision to remove it.
Authors’ response: After re-reading the text, we agreed with the reviewer and removed the text and references. “However, ethnic differences often lead to different oral beliefs and behaviors [12,13])”.
- ‘Considering the negative effects of orthodontic therapy in the first months of treatment and the need to strengthen the available evidence on this issue, the study aimed to investigate the impact of the first six months of fixed orthodontic treatment on the quality of life of adolescents.’ The understandability of the relevant sentence is low and may cause confusion in the reader. My advice to researchers is to divide the sentence and present the purpose of the study clearly to the reader as a separate sentence.
Authors’ response: Part of the sentence has been removed: “Considering the negative effects of orthodontic therapy in the first months of treatment and the need to strengthen the available evidence on this issue,…”
- Please add the null hypothesis sentence after the sentence explaining the purpose of the study and make a statement in the discussion section that you accept or reject your null hypothesis based on your results.
Authors’ response: The null hypothesis of the study was added to the last paragraph of the Introduction section and a statement was made at the beginning of the Discussion section.
RESULTS
- There are too many tables in the results section and it negatively affects the understandability of the article. Researchers do not need to describe all findings one by one. They can highlight important findings and reference others in the table.
Authors’ response: Some results relating to tables 1, 2, 3 and 4 have been removed from the text.
It has been deleted from the 1st paragraph: “and 42.3% of the adolescents’ families had an income less than two times the Brazilian monthly minimum wage.”
It has been deleted from the 2nd paragraph: “...T4 for the “OS” domain, between T0 and T1, T2, T3, and T4 for the “EW” domain, and between T0 and T2, T3, and T4 for the “SW” domain.”
It has been deleted from the 3rd paragraph: “...For the item “mouth sores”, a gradual increase was found in the mean score from one week after bonding, and a reduction was observed at the six-month evaluation....”; “…“concerned about what others think”…”
It has been deleted from the 4th paragraph: “…and a significant change was found regarding the "FL" domain for "malocclusion" (T0 minus T4, p=0.048).”
GENERAL COMMENTS
I believe that the article needs serious editing in terms of writing language. Please have the entire text revised professionally.
Authors’ response: The English language was revised (Text in purple color).
Comments on the Quality of English Language
I believe that the article needs serious editing in terms of writing language. Please have the entire text revised professionally.
Authors’ response: The English language was revised (Text in purple color).
Based on reviewer 1's suggestions, all changes/additions made to the text of the article are in green font.

Round 2
Reviewer 2 Report
Comments and Suggestions for Authors
I have no comments. I suggest that this paper can be accepted for publication.